# Conservation of Three-Dimensional Structure of Lepidoptera and Trichoptera L-Fibroins for 290 Million Years

**DOI:** 10.3390/molecules27185945

**Published:** 2022-09-13

**Authors:** Russell J. Stewart, Paul B. Frandsen, Steffen U. Pauls, Jacqueline Heckenhauer

**Affiliations:** 1Department of Biomedical Engineering, University of Utah, Salt Lake City, UT 84112, USA; 2Department of Plant and Wildlife Sciences, Brigham Young University, Provo, UT 84062, USA; 3LOEWE Centre for Translational Biodiversity Genomics (LOEWE-TBG), 60325 Frankfurt, Germany; 4Senckenberg Research Institute and Natural History Museum Frankfurt, 60325 Frankfurt, Germany; 5Institute for Insect Biotechnology, Justus-Liebig-University, 35392 Gießen, Germany

**Keywords:** AlphaFold2, ColabFold, L-fibroin, silk, caddisflies, silkworms, Trichoptera, Lepidoptera

## Abstract

The divergence of sister orders Trichoptera (caddisflies) and Lepidoptera (moths and butterflies) from a silk-spinning ancestor occurred around 290 million years ago. Trichoptera larvae are mainly aquatic, and Lepidoptera larvae are almost entirely terrestrial—distinct habitats that required molecular adaptation of their silk for deployment in water and air, respectively. The major protein components of their silks are heavy chain and light chain fibroins. In an effort to identify molecular changes in L-fibroins that may have contributed to the divergent use of silk in water and air, we used the ColabFold implementation of AlphaFold2 to predict three-dimensional structures of L-fibroins from both orders. A comparison of the structures revealed that despite the ancient divergence, profoundly different habitats, and low sequence conservation, a novel 10-helix core structure was strongly conserved in L-fibroins from both orders. Previously known intra- and intermolecular disulfide linkages were accurately predicted. Structural variations outside of the core may represent molecular changes that contributed to the evolution of insect silks adapted to water or air. The distributions of electrostatic potential, for example, were not conserved and present distinct order-specific surfaces for potential interactions with or modulation by external factors. Additionally, the interactions of L-fibroins with the H-fibroin C-termini are different for these orders; lepidopteran L-fibroins have N-terminal insertions that are not present in trichopteran L-fibroins, which form an unstructured ribbon in isolation but become part of an intermolecular β-sheet when folded with their corresponding H-fibroin C-termini. The results are an example of protein structure prediction from deep sequence data of understudied proteins made possible by AlphaFold2.

## 1. Introduction

As a general definition, silks are protein-based materials spun into fibers used to create external structures. Silks, silk glands, and spinning apparatuses have originated independently many times across Arthropoda and have contributed to the extraordinary evolutionary success of the phylum [1]. Within the sub-phylum Insecta, silk may have originated independently 23 times in 17 orders [2]. Aquatic caddisflies (Trichoptera) and terrestrial moths and butterflies (Lepidoptera) are sister orders in the Holometabola clade, insects that undergo complete metamorphosis. They produce silk throughout their larval stages from labial silk glands. Trichoptera and Lepidoptera diverged from a silk-spinning ancestor around 290 million years ago [3]. Together they comprise the superorder Amphiesmenoptera.

Many terrestrial moth and butterfly larvae dry spin silk into air to construct cocoons prior to pupation. Most aquatic caddisfly larvae, on the other hand, wet spin silk into water. Caddisflies are divided into two major clades that differ in the type of underwater silken structures they construct [4]. The Annulipalpia and Integripalpia larvae spin silk to construct composite fixed-retreats or portable tubular cases, respectively, with adventitiously gathered building materials, such as stones, sticks, and leaves. Larvae from lineages at the base of Integripalpia primarily spin silk to construct underwater pupal cocoons, protected within composite shelters. The lepidopteran and trichopteran silks are deployed in radically different environments: air with varying degrees of humidity for the lepidopterans, and natural water with dissolved minerals and remnants of organic decay for the trichopterans. Adaptation of terrestrial silk to an aquatic environment (or vice versa) would present several divergent chemical and mechanophysical challenges. It would require evolution of distinct chemistries for interfacial adhesion to substrates (dry versus submerged), distinct chemical mechanisms for rapid solidification of nascent fibers spun into either air or water, and distinct crosslinking interactions to provide sufficient tensile strength, extensibility, and toughness.

Despite the ancient divergence of Lepidoptera and Trichoptera and the extensive adaptations this required, faint molecular fingerprints of the Amphiesmenoptera ancestral silk are visible in the protein sequences of present-day lepidopteran and trichopteran silks. In both orders, the major protein components of the silk fiber core are heavy chain fibroin (H-fib) and light chain fibroin (L-fib). The macroscale structural organization of the H-fibs are similar: blocky, repetitive central regions flanked by short, nonrepetitive N- and C-terminal regions. The central regions show no sequence conservation between the orders, but both contain functionally homologous nano-crystalline domains of stacked β-sheets in the spun fibers that provide tensile strength [5,6,7,8]. After comparing the small number of sequences available at one time, it was reported that the non-repetitive N- and C-termini are conserved between caddisfly and moth H-fibs [9,10]. Sequence conservation in the N-termini was only about 20%, but the positions and spacings of several acidic and basic residues were retained. Sequence conservation in the C-termini was higher, with more than 50% positional conservation of amino acid chemical type (acidic, basic, hydrophobic, etc.). Significantly, all moth and caddisfly H-fib C-termini have a conserved cysteine at approximately –20, which has been shown in *Bombyx mori* to form a disulfide bond with C172 of L-fib [11].

Amphiesmenoptera L-fibroin protein sequence conservation has also been reported. Sequence identities were approximately 25% when three species of moth were compared to three species of caddisfly representing each of the three caddisfly clades [9]. Taking into consideration amino acid type, the overall conservation between moths and caddisfly L-fibs was approximately 50% [10,12]. The cysteine involved in intermolecular crosslinking with H-fib, equivalent to C172 in *B*. *mori*, was conserved in all available L-fib protein sequences. Mutations that disrupt this intermolecular disulfide link in *B. mori* are thought to prevent formation of the H- and L-fib complex in the endoplasmic reticulum and subsequent secretion of the fibroins out of silk gland cells into the silk gland lumen [13,14]. The sequence conservation in both the L-fib and H-fib C-termini between Lepidoptera and Trichoptera implies trichopteran fibroins are also disulfide crosslinked, and covalent complex formation though the H-fib C-terminus may play a similar critical role in the structure, stability, and secretion of the fibroin complex, a role that has been maintained thoughout 290 million years of Lepidoptera and Trichoptera divergence. This conservation is perhaps all the more extraordinary considering that L-fibs are not found in Saturniidae moths [15,16], which is definitive evidence that perfectly functional lepidopteran silks [17] can be produced without L-fibs.

Here, we report comparisons of the three-dimensional (3D) structures of L-fibroins from both orders in an effort to identify higher-order structural changes that may have contributed to the divergent use of silk in water versus air. The 3D structures were generated using the ColabFold [18] implementation of AlphaFold2 [19] in a Google Colaboratory notebook. ColabFold incorporates AlphaFold2 (AF2) with modifications to greatly shorten run times while maintaining the same high accuracy of structure predictions. AF2 is the current state-of-the-art computational approach for protein structure prediction [19,20]. AF2′s deep learning, neural network-based algorithms were trained on experimentally determined protein structures in the Protein Data Bank (PDB). The network predicts 3D structures from a protein’s primary sequence, experimentally derived homologous protein templates if available, and multiple sequence alignments (MSAs) to exploit deep evolutionary histories of homologous proteins. This approach to protein structure prediction has been made possible by the explosive growth of protein sequence databases, resulting from advances in genomic sequencing and annotation. Existing protein 3D templates are not critical for accurate predictions. In the majority of cases, AF2 produces protein structures with accuracies within the margin of error of experimental methodologies, even when no known structural template exists for the target sequence [21]. In effect, AF2 greatly expands the landscape of experimental protein structures and populates it with additional reliable protein structures derived from protein sequence data alone. ColabFold also incorporates AlphaFold2-multimer [22] for predicting the structures of multiprotein complexes, which allowed preliminary explorations of 3D interactions between L-fibs and their corresponding H-fib C-termini.

## 2. Results

### 2.1. L-Fibroin Sequence Alignments

A multiple sequence alignment (MSA) was generated using 33 non-redundant L-fib protein sequences, 22 from Lepidoptera and 11 from Trichoptera (Appendix A). Overall, sequence conservation between lepidopteran and trichopteran L-fibs was low. Not including the secretion signal peptides (SP), there were only 5 (2%) identical positions and 26 (10%) similar positions with conservative substitutions based on the Gonnet PAM250 matrix. Alignment of the 22 lepidopteran sequences separately (not shown) resulted in 11 (3.9%) identical and 46 (16.4%) similar positions. Alignment of the 11 trichopteran sequences separately (not shown) resulted in 61 (23.1%) identical and 67 (25.4%) similar positions. The larger number of protein sequences in the alignment resulted in significantly lower sequence conservation between the orders than previously recognized [9,10].

For clarity of presentation, only the *Bombyx mori* (*Bm*) and *Hesperophylax occidentallis* (*Ho*) alignments, extracted from the full alignment (Appendix A), are shown in Figure 1 as representative lepidopteran and trichopteran sequences. Note that it is not a pairwise alignment. *Bm* is the domesticated silkworm moth. *Ho* is a tube case-making caddisfly (Integripalpia) that prefers building its tubular case with stones. The lepidopteran L-fibs have an 18–22 amino acid N-terminal insertion relative to the caddisfly sequences, and a second 6–10 amino acid insertion at position *Bm*179 (the sequence numbers in Figure 1 include the SP). Caddisflies have a 3–11 amino acid insertion at *Ho*214. The five invariant residues are shaded green, strongly conserved residues dark gray, and less strongly conserved residues light gray. Three of the five identical residues in all 33 sequences are cysteines (C), two of which (*Bm* C101 and *Bm* C160) form an intramolecular disulfide bond in *B*. *mori* L-fib. The third (*Bm* C190) forms an intermolecular disulfide bond to the H-fib C-terminus, as referenced in the introduction.

### 2.2. L-Fibroin Predicted Structures

The tertiary structures of *H. occidentalis* (*Ho*) and *B. mori* (*Bm*) L-fibs, predicted by ColabFold, are presented as representative examples for Trichoptera and Lepidoptera (Figure 2A,B). AlphaFold2 produces per residue accuracy estimates, via the predicted Local Distance Difference Test (pLDDT), as a measure of confidence in the structure prediction. The predicted tertiary structures are color-coded with the ColabFold per residue pLDDT confidence scores. Dark blue regions have >90% confidence. Regions with decreasing confidence range from cyan (80–90%), to green (70–80%), yellows (60–70%), and reds (<50%). Per residue pLDDT plots (Appendix A) show that the confidence is ≥95% in the α-helical regions for both *Ho* and *Bm*. The pLDDT scores dip in positions corresponding primarily to turns and loops connecting the α-helices. Regions with low confidence often correspond to intrinsically disordered structures [23].

AphaFold2 predictions of a protein’s 3D structure are based on their primary amino acid sequences and evolutionary information derived from MSAs of homologous proteins. The accuracy of the predictions therefore depends primarily on the number and diversity of sequences in the MSA. Accuracy decreases substantially with less than 30 sequences and reaches a plateau with little gain beyond ~100 sequences [19]. Sequence coverage and similarity plots for *Ho* and *Bm* (Appendix A) show the L-fib MSAs comprised about 60 sequences. The majority were lepidopteran L-fib sequences, including partial sequences and multiple non-identical sequences from the same organism. A minority were trichopteran L-fibs because far fewer trichopteran sequences have been deposited in UniRef100. This is evident in the per residue sequence identity plots for *Ho* and *Bm* (Appendix A). Nevertheless, the limited sequence coverage was sufficient to create highly accurate 3D structures for both lepidopteran and trichopteran L-fibs. The confidence scores may improve incrementally in the future as more L-fib sequences are added to the databases, but the 3D structures are unlikely to change.

The secondary structures of both *Ho* and *Bm* L-fibs are entirely α-helical. The structure predictions allowed the positions of the helices to be indicated on the linear sequences (Figure 1) by blue and red arrows for *H. occidentalis* and *B. mori*, respectively. The helices are numbered consecutively from the N-termini in Figure 2. A bundle of six anti-parallel helices of similar length, H1-3 and H6–8, lies roughly perpendicular to a bundle of four anti-parallel helices, H4–5 and H9–10. An unstructured loop of 24 amino acids, indicated by yellowish text, connects H5 and H6 (Figure 1). The structures are oriented similarly in Figure 2 to look straight through H1, from C to N. The six-helix bundle is at the top, the four-helix bundle is at the bottom, and the unstructured H5→H6 connector is in front of the four-helix bundle. The N-terminal extension of the *Bm* L-fib is an unstructured ribbon pointing upward from the back of H1.

The *Ho* and *Bm* L-fib structures were superimposed using PyMOL (Figure 2C). The *Ho* structure is gray, the *Bm* structure is blue, and the H5→H6 connectors are orange and red, respectively. The orientation and relative positions of the helices are nearly the same in the two structures. The trichopteran 3–11 amino acid insertion between H9 and H10 (Appendix A) extends the H10 helix on the N side, resulting in a pronounced protuberance on the lower left side of the *Ho* structure relative to the *Bm* structure (Figure 2). For inspection of other orientations, the structures can be rotated in 360 degrees around two axes using Supplemental Video 1 (SV1). ColabFold predicted structures of additional L-fibs from both orders were similarly superimposable (not shown).

The 3D spatial locations of the conserved amino acids of *Ho* and *Bm* L-fibs are shown in Figure 3. A strong correlation between AlphaFold2-predicted backbone accuracy and sidechain accuracy was reported [19] Therefore, the high backbone pLDDT scores for the L-fibs suggest that the sidechain orientations are also likely to be accurate. The five identical residues (magenta) were labeled and numbered (the numbers correspond to the sequences without the signal peptide). The two cysteines (C) that form the intramolecular disulfide bond connect near the center of the H5→H6 connector to the beginning of H4. The C that forms the intermolecular disulfide bond with the H-fib C-terminus is in the center of H7, oriented toward the surface of the protein. The invariant arginine (R), oriented toward the surface, also occurs in the unstructured H5→H6 connector about halfway between the disulfide bond and the beginning of H4. The invariant tryptophan (W) is buried between H6, 7, and 8. The strongly conserved amino acids (green) are clustered around the invariant residues, many of which appear to contribute to the hydrophobic core of the protein. The positively charged R or lysine (K) on H7 and the negatively charged glutamic acid (E) on H6 may contribute to the stability of the structure though electrostatic interactions.

### 2.3. Electrostatic Surface Potentials

The electrostatic (ES) potentials of the ColabFold structures at the solvent accessible surface are shown in Figure 4. Although the 10-helix structure is well conserved, the surface ES potential is not conserved between Lepidoptera and Trichoptera. The initial structures (Figure 4A,E) are presented in the same orientation as in Figure 3, looking straight through H6. The protuberance due to the H10 extension in *Ho* is apparent at the bottom left of the structure. The initial *Bm* structure (E, front) has a patch of negative potential at the base of the N-terminal extension that is less pronounced in the *Ho* structure. The *Ho* structure, on the other hand, has a negative patch on the right side of the molecule that is not present in *Bm*. Rotating the molecules 90° ccw around the horizontal access (Figure 4B,F, bottom) reveals a patch of positive potential associated with the *Ho* H10 extension (sequence insertion), which contains three Ks and an R (Figure 1). The corresponding area of *Bm* is relatively neutral. Rotating another 90° ccw (Figure 4C,G, back) reveals a patch of negative potential on *Ho* without a corresponding negative patch on *Bm*. Finally, rotating another 90° ccw (Figure 4D,H, top) again shows the patch of negative potential at the base of the *Bm* N-terminal extension. The corresponding surface of *Ho* is neutral. For examination of additional orientations, the structures for *Ho* and *Bm* were rotated by 360 degrees around two axes in Supplemental Videos 2 and 3, respectively (SV2, SV3).

### 2.4. Heteroduplex Structures

ColabFold includes an AlphaFold2 multimer for predicting homo- and heteromeric structures [18,22]. L-fibs from four species, two trichopteran and two lepidopteran, were folded with 40 amino acids from the H-fib C-terminus of the same species (Figure 5). The structures are oriented as before, looking through H6. The H-fib C-termini contact the L-fib in similar ways in all four species, wrapping over the top from the left side. Intriguingly, the N-terminal extensions of the two lepidopteran L-fibs, *B*. *mori* (Figure 5C) and *G*. *mellonella,* form intermolecular anti-parallel β-strands with the H-fib C-termini (Figure 5D). They then wrap over the top to contact H7. The two trichopteran C-termini, on the other hand, contact the L-fib near the C-end of H1 on the left side similarly to the lepidopterans, and wrap over the top of H7, but then make a sharp turn between H6 and H7. The trichopteran H-fib C-termini then follow a course on the L-fib surface underlaid with the most strongly conserved amino acids (Figure 6). In all four structures, the conserved C-20 (numbered from the end of the H-fib C-termini) are positioned in close proximity to the invariant C projecting upward from H7 in the L-fibs.

The confidence in the L-fib folded chains in the heterocomplex was mostly high (dark blue). Despite low sequence coverage (less than 20 H-fib C-termini), the pLDDT scores for the C-termini chains in the complexes were 70–80% (green) in several short stretches and >90% (blue) for the β-strands formed with the N-terminus of *G*. *mellonella*. In future work, custom MSAs with increased depth and diversity may further improve prediction accuracies of the heterocomplexes. To access the full chain accuracy of predicted structures, AlphaFold 2 outputs pTM (predicted template modeling) scores on a scale 0 to 1 as a global superposition metric. AlphaFold-multimer additionally outputs ipTM, a metric similar to pTM that was modified to score the accuracy of the interface between different chains in a complex. The scores for the heterocomplexes are listed in Table 1. The ipTM for the *G*. *mellonella* heterocomplex reached the confidence range.

## 3. Discussion

Amphiesmenoptera L-fibroin is a protein family unto itself, apparently an invention of the common silk spinning ancestor of Lepidoptera and Trichoptera. Linear sequence homology searches returned only L-fibs from Lepidoptera and Trichoptera; no distant non-Amphiesmenoptera homologs were identified. A search for similar 3D protein structures using VAST (Vector Alignment Search Tool, NCBI) with the L-fib 3D coordinates from ColabFold returned no homologous structures—the 10-helix fold of L-fib is so far unique amongst experimentally determined protein structures.

After ~290 million years of divergence, only 5 L-fib amino acid positions (~2%) were invariant, tolerating no amino acid substitutions, and only 26 amino acid positions (~10%) tolerated chemically conservative substitutions in the 33 aligned sequences. The vast majority of sequence positions were highly tolerant of amino acid substitutions. Large volumes of the L-fib structures have no conserved amino acids. Helices 1, 2, and 9 have no conserved amino acids, yet they persist in the core. Some of the sequence drift may be due to coevolution, or compensating mutations, that are difficult to ascertain by eye in the alignments. Despite the overall low conservation of the protein sequence, the core tertiary structure of L-fib, the novel 10-helix fold, was highly conserved.

The small number of conserved positions point to their outsized importance for the initial folding of the structure, or stabilization of the final structure, or direct interactions with co-actors in L-fib’s function. Three of the invariant amino acid positions are cysteines. There are no substitutes for paired cysteines if covalent disulfide bonds are required for stability or function. Two of the cysteines form an internal disulfide bond that may rigidize the tertiary structure, particularly the H5→H6 loop, covalently coupling it to the beginning of H4. The third covalently couples H7 to H-fib. The remaining strongly conserved amino acids are mostly clustered on the internal interface of H6, H7, and H8. Fourteen of the 30 conserved amino acids (47%) are on H6, H7, and H8, which together comprise about 15% of the sequence. The invariant tryptophan (W) and a highly conserved phenylalaine (F) are buried at the interface of H6, H7, and H8. These bulky, hydrophobic, and aromatic amino acids may be critical to establishing precise spacing between the helices. It appears the role of most of the conserved residues may be to establish and stabilize the core structure, perhaps to maintain the precise 3D orientation of H7 for its critical interaction with the H-fib C-terminus.

Although the organization of the core helix bundles is highly conserved in lepidopteran and trichopteran L-fibs, there are intriguing variations in the 3D structures that hint at divergent intermolecular interactions. These variations may reflect some of the molecular evolution of their silk proteins that accompanied divergence of Lepidoptera and Trichoptera into distinct habitats. First and most conspicuous, the N-terminal L-fib extension is an unstructured ribbon in the isolated lepidopteran structures. When folded in the presence of the H-fib C-terminus, it becomes structured, forming a β-strand in anti-parallel association with a β-strand of the C-terminus (Figure 5). The association through a secondary structure may further strengthen the interaction between Lepidoptera H- and L-fibs compared to Trichoptera, which lack the N-terminal extension. Second, there are length variations in several of the helices, resulting in volumetric shape differences (Figure 4). Trichopteran H10 is extended on both ends relative to Lepidoptera (SV1). Additionally, trichopteran H6 and H7 are slightly shorter than lepidopteran H6 and H7 (SV1). There is also length variation in the H6 to H7 loop; the Lepidoptera have variable length insertions relative to Trichoptera (Appendix A). H6 and H7 form an interesting territory on the L-fib surface; the trichopteran H-fib C-termini take a sharp turn between H7 and H6 to follow a different path than the lepidopteran C-termini, a path underlaid by the most strongly conserved amino acids (Figure 6). Third, the patchworks of electrostatic surface potentials are distinct. In general, the disparate distributions may point to interactions with order-specific external co-actors. For example, the bottom surface of the H10 extension of *Ho* corresponds to a patch of concentrated positive potential in what appears to be a shallow pocket (Figure 4B), perhaps the hallmarks of a binding site for an unknown protein or other external factor. The region at the base of the lepidopteran N-terminal extension is distinctly more negative than the corresponding region in Trichoptera, which may be important for the formation of the intermolecular β-sheet connection to H-fib that is unique to the Lepidoptera.

The function of L-fib has been most studied in *B*. *mori,* in which the covalently linked H-fib and L-fib heterodimers associate non-covalently with the P25 glycoprotein (aka fibrohexamerin) in a 6:6:1 complex, referred to as the elementary unit (EU) of silk [24,25]. The P25 protein or a recognizable homolog has not been found in Trichoptera silks [26]. Analysis of *B. mori* L-fib mutations that disrupt the intermolecular disulfide link with H-fib suggested the formation of the H- and L-fib covalent complex in the endoplasmic reticulum was necessary for the subsequent secretion out of silk gland cells into the silk gland lumen [13,14]. In other words, L-fib functioned similarly to a chaperonin during fibroin synthesis, although it is a component of the final fibers. The EUs cluster into larger nanocomplexes as the storage form of silk in the silk gland lumen. The role of ion concentration and pH gradients in transition of the storage form of silk into silk fibers during transit through the silk gland has been well documented in *B. mori* [27,28,29,30,31]. In this context, the patchy electrostatic surface potential of L-fib could fluctuate in response to changes in pH, depending on the pKa of the underlying amino acids. Likewise, areas of strong surface potential could be coordination sites for mulitivalent ions, or subject to shielding by monovalent ions to modulate interactions with other macromolecules, including H-fib. Given L-fibs’ participation in the structure of the EUs in *B. mori*, and the possibility that patches of strong surface potential could be interaction sites with external factors, it is at least conceivable that modulation of L-fib surface potentials by silk gland pH and ion concentration gradients could contribute to the molecular rearrangements that occur during the transition of the storage form to the fiber forms of silk proteins. The same may be true in Trichoptera, in which silk fiber precursors are also stored as complex fluids of nanometer-scale complexes in the posterior silk gland and transition into fibrous structures on passage through the silk gland [32,33,34]. Gradients in silk gland pH or ion concentrations have not been reported in Trichoptera to our knowledge.

Do homologous L-fib 3D structures imply homologous function in Lepidoptera and Trichoptera? There have been no direct studies on the structural role of L-fib in caddisfly silk elaboration. That L-fib is likely covalently coupled to H-fib and may therefore also act as a secretion chaperonin may be the full extent of inference currently possible from comparisons to lepidopteran L-fibs. The absence of L-fib in saturniid moths demonstrates that workarounds are possible, which suggests that even though Trichoptera conserved the L-fib core structure and covalent linkage to H-fib, details of its function(s) may have diverged dramatically, including adjustments necessary for spinning silk into water versus air. The sites of structural variations in the conserved core structure described above may point to important targets for further experimental exploration of the evolution of L-fibroin functions in Lepidopteran and Trichoptera.

The substantial commercial and technological interest in insect silks, so far focused almost exclusively on *B. mori* silkworm silk, provides further motivation for investigating L-fibroin’s structural role(s). *B. mori* silkworm silk has been of enormous commercial interest for millennia as the basis of the silk textile industry. In this context, the limited comparisons of *B. mori* with other lepidopteran L-fibroins reported here suggest the L-fibroin 3D structure was highly conserved during a few thousand years of domestication and intense artificial selection for efficient, high-volume production of textile silks. It is possible that more extensive and fine-toothed comparisons of *B. mori* L-fibroin 3D structure with other lepidopteran wildtype L-fibroin 3D structures may reveal subtle structural variations resulting from artificial selection pressures.

Beyond textiles, there has been considerable interest in technological applications of *B. mori* silk, particularly as biomedical materials. The history and breath of applications of silkworm silk in medicine has been thoroughly reviewed by Holland et al. [35]. Technological applications are based on regenerated silkworm fibroin derived by degumming and solubilization of natural silk fibers. Stoichiometrically, L-fibroin is an equal one-to-one partner with H-fibroin in naturally spun fibers, although by mass it comprises a small fraction of the fiber. The absence of a reducing step in standard silkworm silk regeneration protocols suggests that L-fibroin remains covalently coupled to H-fibroin through a disulfide bond in regenerated silk fibroin. The presence or absence of L-fibroin is rarely mentioned in the regenerated silk fibroin literature [36]. The use of regenerated silk fibroin to artificially create fibers with the material properties of natural silkworm silk proved challenging and required far from natural processing steps [37]. Considering its possible function as a chaperonin-like molecule during the secretion of H-fibroin, the potentially irreversible loss of L-fibroin structure during silk fiber solubilization could contribute to these challenges.

Technological applications of caddisfly silk remain aspirational. As a silk naturally selected to function as both an adhesive, and as a strong, highly extensible, and tough fiber while fully submerged in water [38,39], caddisfly silk is a logical model for the design of materials intended to be used in a fully hydrated environment—living tissue, for example. As the silk technology community makes progress in exploiting the unique properties of caddisfly and other insect silks, the contribution of L-fibroin to silk’s structure and material properties may be a useful consideration. This may be especially true in efforts to use structural elements and/or production methods of different silk systems to create hybrid silk materials.

## 4. Materials and Methods

### 4.1. L-Fibroin Alignments

A search of UniProtKB for L-fib returned 54 sequences. After removing spurious and species-redundant sequences, e.g., there were 12 non-identical *B*. *mori* sequences. The remaining 22 lepidopteran sequences were aligned with 11 trichopteran L-fib sequences using Clustal Omega [40] on UniProt with default settings (Appendix A). Secretion signal peptides were included in the alignments. The lepidopteran and trichopteran sequences were also aligned as separate groups. See Appendix A for additional details regarding trichopteran L-fibroin sequences.

### 4.2. Structure Predictions

Signal peptides were predicted with SignalP 6.0 [41] and removed from the sequences before structure predictions. Therefore, the amino acid numbers in the sequence alignments and the predicted structures differ by the length of the respective signal peptide.

L-fib structures were generated using ColabFold [18] in the Google Colaboratory environment (https://colab.research.google.com, accessed on 15 August 2022). ColabFold uses AlphaFold2 [19] and Alphafold2-multimer [22] to predict the structures of individual proteins and protein complexes. AlphaFold L-fib structures have been generated for *B*. *mori* (P21828 (FIBL_BOMMO)) and *G*. *mellonella* (Q26427 (FIBL_GALME)) in an automated pipeline and are accessible on UniProt, and in the AlphaFold Protein Structure Database [42], thanks to the EMBL—European Bioinformatics Institute (https://alphafold.ebi.ac.uk, accessed on 15 August 2022).

ColabFold has advanced options. Template mode was consistently selected as PDB70, although no structural templates were found for any of the sequences we folded. A structural template is not required for accurate structure prediction. The multiple sequence alignment (MSA) input is a more critical factor in prediction accuracy. For MSA options, MMseqs2 (UniRef+Environmental) and paired+unpaired were used. For model type, the auto option was used, which automatically detects if more than one sequence is entered, and if so, will use Alphafold2-multimer. The default recycle number is 3. We found that in some cases increasing the recycle number resulted in incremental improvements in the intrinsic accuracy metrics, e.g., pLDDT. Heteroduplex (L-fib with H-fib C-termini) predictions were run with 24 recycles.

PDB structure files output by ColabFold were visualized and manipulated in Pymol (The PyMOL Molecular Graphics System, Version 2.0, Schrödinger, LLC). Supplemental videos were generated in Pymol. Electrostatic surface potentials were generated using the APBS (Adaptive Poisson-Boltzman Solver) Pymol plug-in. The molecules were prepared using pdb2pqr. The electrostatic potential at the solvent accessible surface is shown in Figure 4 and in supplementary videos 2 and 3—red for negative and blue for positive.

## 5. Conclusions

AlphaFold2 generated convincing 3D structures, even though Amphiesmenoptera L-fibroins occupy a comparatively barren region of the experimental protein sequence/structure landscape. We bore in mind that these are algorithm-generated structures. Nevertheless, on the whole, we found the results compelling, in large part because of the extensive benchmarking by others of the accuracy of AF2 and AF2-multimer, including the intrinsic confidence scoring metrics [20,21,43,44]. We found that despite low protein sequence conservation, a 10-helix core structure was strongly conserved, but with intriguing order-specific variations. The consistency in both the similarities and differences between and within Trichoptera and Lepidoptera L-fibs seems more than coincidental. The predicted confirmation of known structural features, such as the intra- and intermolecular disulfide bonds, and emergent details, such as the Lepidoptera-specific N-terminal extension forming an intermolecular β-sheet with the H-fib C-termini, and the distinct distributions of surface electrostatic potential, add credence to the models. Finally, the results point toward the remarkable opportunities created by AF2 that may lie ahead for predicting protein structure from sequence data alone, by exploiting deep evolutionary histories, especially for proteins such as silk L-fibroins that have received little experimental attention, but for which genomic sequencing efforts are creating treasure troves of evolutionary information.

## Figures and Tables

**Figure 1 molecules-27-05945-f001:**
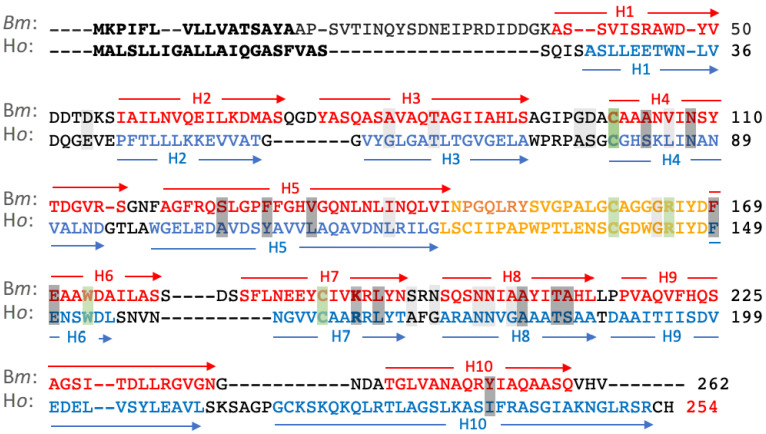
Alignment of *B. mori* (*Bm*) and *H*. *occidentalis* (*Ho*) L-fib sequences extracted from the full sequence alignment (Appendix A). This is not a pairwise alignment of *Bm* and *Ho*. Secretion signal peptides are bold and are included in the numbering. Positions of helices are indicated by colored text and arrows. The unstructured region connecting helix 5 and 6 is indicated by yellow text. Invariant amino acids are shaded green. Strongly conserved amino acids and less strongly conserved amino acids are shaded dark gray and light gray, respectively.

**Figure 2 molecules-27-05945-f002:**
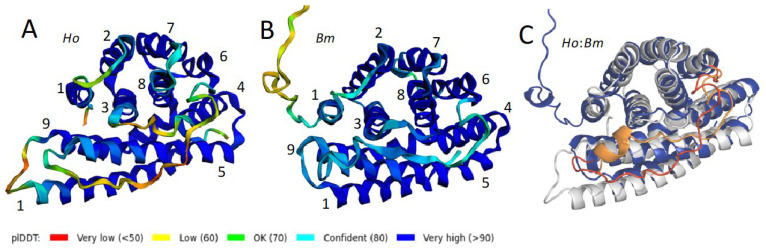
ColabFold predicted L-fib structures color-coded by pLDDT. Helices are numbered. (**A**) *H. occidentalis* (*Ho*). (**B**) *B. mori* (*Bm*). The helices have been numbered consecutively from the N-termini. (**C**) Overlay of *Ho* (gray) and *Bm* (blue) ColabFold structures. The unstructured loops connecting H5 and H6 are shown in orange (*Bm*) and red (*Ho*) for better visibility. See Supplemental Video 1 (SV1) to examine more orientations of the overlaid structures.

**Figure 3 molecules-27-05945-f003:**
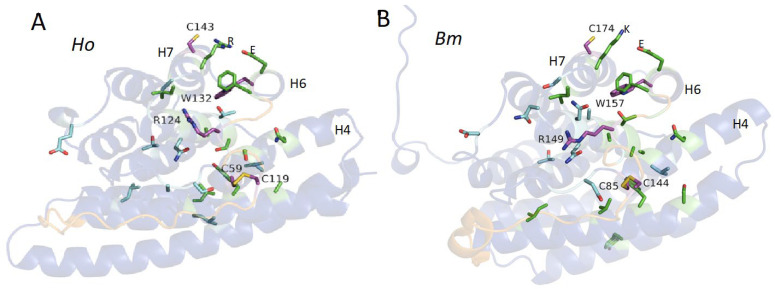
Locations of conserved amino acid sidechains. Both structures are oriented to look through H6. (**A**) *H*. *occidentalis* (*Ho*) (**B**) *B. mori* (*Bm*). Identical amino acids (magenta) are labelled with amino acid number. Strongly conserved amino acids (green) and conserved amino acids (cyan). The H5→H6 loop is orange. The numbers exclude the SP. H4, H6, and H7 are labelled.

**Figure 4 molecules-27-05945-f004:**
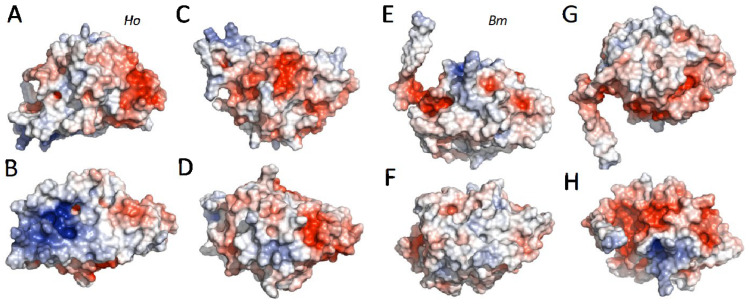
Electrostatic surface potentials. (**A**–**D**) *H*. *occidentallis* (*Ho*). (**E**–**H**) *B. mori* (*Bm*). The initial structures (**A**,**E**) are oriented the same as in Figure 3. Each successive structure has been rotated 90° counter-clockwise around the horizontal axis: (**A**,**E**) front, (**B**,**F**) bottom, (**C**,**G**) back, (**D**,**H**) top. Blue = positive potential. Red = negative potential. See Supplemental Videos SV2 and SV3 for more orientations.

**Figure 5 molecules-27-05945-f005:**
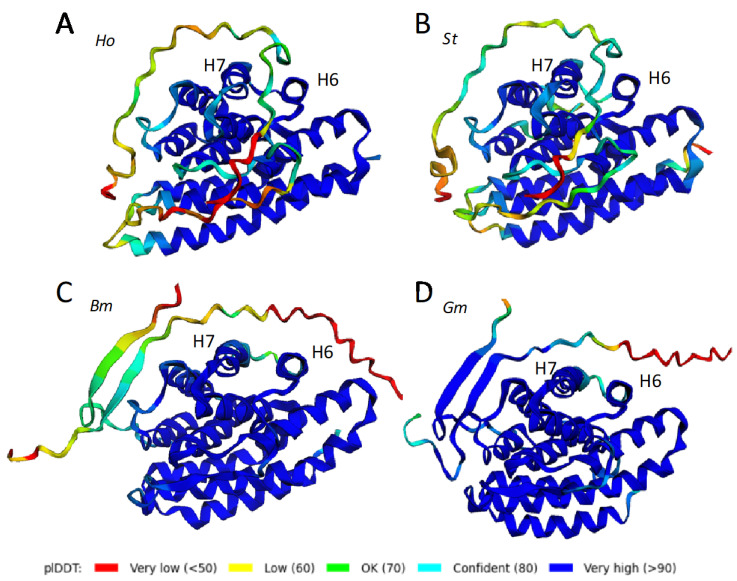
Heteroduplex structures: L-fibs with corresponding H-fib C-termini from the same species. (**A**) *H*. *occidentallis* (*Ho*). (**B**) *S. tienmushanensis* (*St*). (**C**) *B*. *mori* (*Bm*). (**D**) *G*. *mellonella* (*Gm*). The chains were colored using the Colabfold-generated per residue pLDDT confidence score.

**Figure 6 molecules-27-05945-f006:**
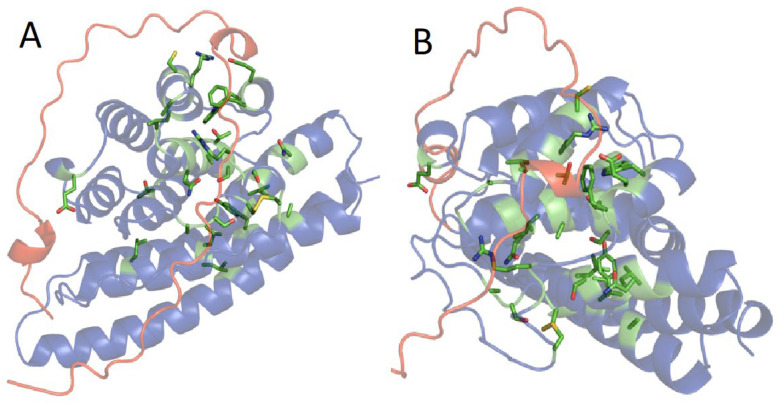
Heterocomplex of L-fib (blue) and H-fib C-terminus (red) of *H*. *occidentallis*. (**A**) Front, looking through H6 as before. (**B**) Topview. All conserved amino acids were colored green to highlight clustering.

**Table 1 molecules-27-05945-t001:** Alphafold-multimer scores.

	Ave. pLDDT	pTM	ipTM
*H. occidentalis*	85.3	0.86	0.58
*S. tienmushanensis*	85.2	0.87	0.61
*B. mori*	84.8	0.87	0.62
*G. mellonella*	90.3	0.90	0.83

## Data Availability

Newly reported l-fibroin sequences (see Appendix A) can be found on figshare: https://doi.org/10.6084/m9.figshare.20240847.v1 (accessed on 15 August 2022).

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
