# Peer review of "Conservation of Three-Dimensional Structure of Lepidoptera and Trichoptera L-Fibroins for 290 Million Years"

_molecules, 2022, doi:10.3390/molecules27185945_

Round 1

Reviewer 1 Report

The research objective(s) and methodology are clearly described. The conclusions drawn from the comparison of the Trichoptera and Lepidoptera structures of L-fibroins generated by AlphaFold2 are sound. The authors have paid due attention to the caveats regarding the generation of three-dimensional models by AlphaFold2.

Author Response

We appreciate the reviewers positive comments.

Reviewer 2 Report

General

Stewart et al. studied molecular changes in L-fibroins in the sister orders Trichoptera and Lepidoptera which diverged 290 years ago.  Using the AlphaFold2 tool they predicted 3-D structurers of L-fibroins from representative samples stemming from both orders.  They found, that despite the ancient divergence and different habitats, a “10-helix core structure was strongly conserved in L-fibroins in both orders”.  The authors dwell on this and sought to better understand the evolution of silks adapted to water and air.  I note that I was unable to open the supplemental video as I do not have the needed software.  Take this into account as many readers will face the same problem (or al least so I assume).

The manuscript addresses a novel and pertinent topic, is very well written, and has no fundamental flaws.  Although the topic may sound to be of narrow interest geared to a specialized audience, the findings have broad applications and will interest a broad audience, all the way from evolutionary biologists to engineers interested in biomimicry for the development of resistant fibers usable in contrasting environments (air vs water).  In this respect, I suggest adding a paragraph in the discussion dwelling on the applied implications of the findings.

Given that Molecules is a general-interest journal, it would be perhaps useful to the readership to frame the introduction more broadly, providing some generalities on insects, their broad evolution, and the overall phylogenetic placing of Trichptera and Lepidoptera.  Are they basal or derived groups?  Within Insecta, are they the only ones producing silk?  All this will help the reader to understand the rest of the story.  Cite for example general books such as “Evolution of the Insects” by David Grimaldi and Michael S. Engel (2005).  In other words, make your introduction more reader friendly.  Perhaps a picture of representatives of each groups and a cocoon (air vs aquatic) in the initial part of the manuscript would be appropriate, illustrating with a close-up, how the silk looks like.

I also suggest a paragraph in the introduction dwelling on the synthesis of L-fibroins to allow the reader to gain a broader picture on the topic being addressed.

Based on this comment “Overall, sequence conservation between lepidopteran and trichopteran L-fibs was low”, I wonder if it would have been useful to compare, within both groups, basal vs derived species.  I understand that this would have likely not changed the robust findings being reported, but it would have been interesting.

Expand on this finding: “Although the 10-helix structure is well conserved, the surface ES potential is not conserved between Lepidoptera and Trichoptera”.

Expand on this finding and dwell on its implications: “Three of the invariant amino acid positions are cysteines”.  What role do cysteines play in the synthesis or structural stability of L-fibroins?

You stated initially that you sought to better understand the evolution of silks adapted to water and air.  Need to dwell on this in more detail in the discussion.

Specific

Line 117. The larger number of protein sequences …

Line 122. … in Figure 1 (capitalize F).  Applies throughout ms.

Line 128.  … The five invariant … Applies elsewhere to numbers under 10 which should be spelled out.

Line 186. Doesn’t the period go after the parenthesis?  Applies elsewhere.

Line 261.  Say who that “common silk-spinning ancestor was.

Lines 417 – 423.  Need to add the missing information as only R.S. is identified.

Author Response

We appreciate that the reviewer found the results interesting and of broad interest.  We thank the reviewer for their several useful suggestions to improve the manuscript.  

General
Stewart et al. studied molecular changes in L-fibroins in the sister orders Trichoptera and Lepidoptera which diverged 290 years ago.  Using the AlphaFold2 tool they predicted 3-D structurers of L-fibroins from representative samples stemming from both orders.  They found, that despite the ancient divergence and different habitats, a “10-helix core structure was strongly conserved in L-fibroins in both orders”.  The authors dwell on this and sought to better understand the evolution of silks adapted to water and air.  I note that I was unable to open the supplemental video as I do not have the needed software.  Take this into account as many readers will face the same problem (or al least so I assume).

The videos are mpg formatted. This is a common format that most video players, Mac or Windows, will recognize. There may have been some problem other than format with transferring the videos.  

The manuscript addresses a novel and pertinent topic, is very well written, and has no fundamental flaws.  Although the topic may sound to be of narrow interest geared to a specialized audience, the findings have broad applications and will interest a broad audience, all the way from evolutionary biologists to engineers interested in biomimicry for the development of resistant fibers usable in contrasting environments (air vs water).  In this respect, I suggest adding a paragraph in the discussion dwelling on the applied implications of the findings.

This is an excellent suggestion.  A section (three paragraphs) on the potential technological implications of the work has been added at the end of the discussion.

Given that Molecules is a general-interest journal, it would be perhaps useful to the readership to frame the introduction more broadly, providing some generalities on insects, their broad evolution, and the overall phylogenetic placing of Trichptera and Lepidoptera.  Are they basal or derived groups?  Within Insecta, are they the only ones producing silk?  All this will help the reader to understand the rest of the story.  Cite for example general books such as “Evolution of the Insects” by David Grimaldi and Michael S. Engel (2005).  In other words, make your introduction more reader friendly.  Perhaps a picture of representatives of each groups and a cocoon (air vs aquatic) in the initial part of the manuscript would be appropriate, illustrating with a close-up, how the silk looks like.

A paragraph was added at the beginning of the introduction to place caddisflies and moths within the broader silk-spinning arthropod phylogeny.  We also find the evolutionary aspects of the results intriguing, but the main theme of the paper is biochemistry and molecular structure of L-fibroin.  Indeed, the paper is in the molecular structure section of "Molecules". 

I also suggest a paragraph in the introduction dwelling on the synthesis of L-fibroins to allow the reader to gain a broader picture on the topic being addressed.

We are not sure of the reviewers meaning.  L-fibroin itself is synthesized like any other protein. Beyond that we described that L-fibroin is covalently bound through cysteines and co-secreted with H-fibroin and may have a chaperonin-like function—lines 86-93 of the introduction. 

Based on this comment “Overall, sequence conservation between lepidopteran and trichopteran L-fibs was low”, I wonder if it would have been useful to compare, within both groups, basal vs derived species.  I understand that this would have likely not changed the robust findings being reported, but it would have been interesting.

This is a good suggestion that may be addressed in future work more focused on the evolutionary aspects of the structure conservation.

Expand on this finding: “Although the 10-helix structure is well conserved, the surface ES potential is not conserved between Lepidoptera and Trichoptera”.

This was addressed in lines 353-366 of the discussion.  We speculated that the distinct patterns of surface potential may influence interactions with other factors in response to ion and pH gradients on transit through the silk gland.  We are reluctant to speculate further.

Expand on this finding and dwell on its implications: “Three of the invariant amino acid positions are cysteines”.  What role do cysteines play in the synthesis or structural stability of L-fibroins?

This was addressed in lines 301-305 of the discussion.  Pairs of cysteines form covalent crosslinks.  We suggested that the internal disulfide bonds may rigidize the structure.  The other cysteine covalently couples L-fibroin to a cysteine in H-fibroin, as described in the introduction, lines 86-93.  We are reluctant to speculate further. 

In the new discussion paragraphs related to technological implications, we point out that the disulfide bond between L-fibroin and H-fibroin suggests that L-fibroin is likely a component in regenerated silk proteins, the presence of which is generally not taken into account in the regenerated silk literature.

You stated initially that you sought to better understand the evolution of silks adapted to water and air.  Need to dwell on this in more detail in the discussion.

It is implicit that there are no firm conclusions that can be drawn with respect to air vs water adaptation. Any further discussion would be too speculative. 

Specific
Line 117. The larger number of protein sequences …
Line 122. … in Figure 1 (capitalize F).  Applies throughout ms.
Line 128.  … The five invariant … Applies elsewhere to numbers under 10 which should be spelled out.

The above specifics were corrected.

Line 186. Doesn’t the period go after the parenthesis?  Applies elsewhere.

The parentheses were removed.

Line 261 (281).  Say who that “common silk-spinning ancestor“ was.

The common ancestor is unknown.

Lines 417 – 423.  Need to add the missing information as only R.S. is identified.

Author contributions have been completed.